# Translation and Validation of the Arabic Version of the Athlete Sleep Screening Questionnaire

**DOI:** 10.3390/healthcare11101501

**Published:** 2023-05-22

**Authors:** Ahmed S. Alhowimel, Aqeel M. Alenazi, Mohammed M. Alshehri, Bader A. Alqahtani, Abdulaziz Al-Jamaan, Faris Alodaibi, Yasir S. Alshehri, Jonathan Charest

**Affiliations:** 1Department of Health and Rehabilitation Sciences, Prince Sattam Bin Abdulaziz University, Alkharj 11942, Saudi Arabia; 2Physical Therapy Department, Jazan University, Jazan 45142, Saudi Arabia; 3Department of Rehabilitation Science, King Saud University, Riyadh 11451, Saudi Arabia; 4Department of Physical Therapy, College of Medical Rehabilitation Sciences, Taibah University, Madinah 41411, Saudi Arabia; 5Faculty of Kinesiology, University of Calgary, Calgary, AB T2N 1N4, Canada; 6Centre for Sleep & Human Performance, Calgary, AB T2X 3V4, Canada

**Keywords:** sleep, screening, athletes, sport

## Abstract

Sleep improves the cognitive and physical performance of athletes. A detailed questionnaire that detects sleep disruptions is required to identify sleep-deprived athletes. This study evaluates the translated Athlete Sleep Screening Questionnaire (ASSQ), a tool suggested by the International Olympic Committee, among Arabic-speaking athletes. The ASSQ was translated into Arabic and examined for floor or ceiling effects, internal consistency, and validity among Arabic-speaking athletes. The Arabic Pittsburgh Sleep Quality Index (PSQI) was employed to assess convergent validity. Ninety athletes (28.9% women) participated and completed this study’s questionnaires. The Cronbach’s alpha for the ASSQ-Sleep Difficulty Score (SDS) was 0.435, and that of the ASSQ-chronotype was 0.632. The SDS and chronotype subset of the ASSQ demonstrated excellent test-retest reliability, with intraclass correlation coefficients of 0.84 and 0.938, respectively. The ASSQ-SDS correlated positively with the PSQI (0.734, *p* = 0.001). The ASSQ-chronotype was inversely associated with the PSQI (*p* = 0.001). This study’s findings can assist clinicians in assessing sleep disorders in sports. The Arabic version of the ASSQ has satisfactory psychometric qualities and can identify clinically relevant sleep problems in athletes.

## 1. Introduction

Sleep is important for athletes’ sports skills, emotional control, and physical and mental well-being. Athletes who do not obtain enough sleep may be more at risk of injury or illness. Sports research has recently focused on how sleep affects athletic performance and recovery [1]. Athletes and coaches have always thought that sleep was the best way to recover [2]; it differs from common therapeutic techniques such as cryotherapy, compression clothing, or electrical stimulation [3,4].

During sleep, anabolic metabolism speeds up [5,6], procedural memory improves [7], and immune responses are boosted [5]. Sleep deprivation or insufficient sleep may significantly affect performance, motivation, effort perception, cognition, and other biological systems [8]. Sleep is also linked to several bodily processes that may help athletes recover from training and competition and adjust to them [9,10,11,12]. Other studies have shown that sleep is important for metabolic homeostasis [13]. Humans’ ability to deal with physical and mental stressors is a key factor in their sports performance [14]. Experience, fitness, motivation, and the typical 24-h changes in physiological and behavioral processes can all impact performance [15].

Although humans are susceptible to changes in their everyday environment [16,17], especially the light-dark cycle [18], the suprachiasmatic nucleus cannot always maintain control over these patterns. Endogenous circadian rhythms and normal sleep–wake cycles can become out of sync [16,19] when athletes are exposed to external disturbances such as nighttime training or competitions and travel. These changes in how people sleep could increase homeostatic pressure and affect emotional control, core temperature, and melatonin levels in the blood, resulting in longer sleep onset latency periods [20]. However, insufficient sleep is likely to lead to lower neurocognitive and physical performance [9,21,22,23]. New research shows that athletes and healthy people sleep for different periods and in different ways. Moreover, athletes are often exposed to factors that affect how long and how well they sleep, such as jet lag, unfamiliar locations, nocturnal training, competitions, or underlying fatigue [24].

Though sleep monitoring has become common in sports, finding athletes who might need help with sleep problems may be helpful. Therefore, it is crucial to identify the individual sleep patterns of athletes and provide them with appropriate education on good sleep hygiene practices to promote optimal sleep quality and duration. Noninvasive and effective interventions, such as actigraphy and other alternatives to polysomnography, can provide important information about sleep and wakefulness during the sports season.

Despite substantial scientific research on the duration and quality of sleep in people of different ages, there is insufficient research on how athletes sleep. However, sleep is known to be one of the most important parts of overcoming fatigue and a critical factor in athletic performance. Therefore, athletes, coaches, and specialists must be aware of factors affecting sleep and the interventions that can be used to track how long and how well an athlete sleeps. These critical measures can be controlled to improve the athlete’s health and performance.

## 2. Background

Sleep is an essential biological mechanism for recovering from high-performance sports’ mental and physical strain [25,26]. Moreover, it plays a crucial role in the recovery of an athlete’s physiological, psychological, musculoskeletal, immune, metabolic, and endocrine systems; sleep is linked to better performance and stress adaptation in athletes [27]. Sleep quality is enhanced by sleep continuity and efficiency; however, the roles of sleep architecture and naps are less clear. Athletes may require more sleep for adequate recovery and adaptation, but no specific guidelines exist for their sleep duration or quality [3,28].

Athletes’ sleep is influenced by several sport-related factors, such as training intensity and travel associated with competitive sports. The challenges to athlete sleep likely vary across different sports due to specificity, culture, and environmental factors. Sleep problems are prominent in contact, combat, and aesthetic sports, possibly due to factors such as concussion history and anxiety. Low energy might affect sleep patterns in some athletes, while individual sport athletes tend to have less sleep than team sport athletes due to earlier training times [28,29,30]. Chronotype matching with sport-specific training schedules could impact an athlete’s success. Sleep difficulties are more noticeable in individual sports during high-risk periods such as during competitions [28,29,30]. Several non-sport factors, such as illness, sleep debt, sex, and stress, can affect athletes’ sleep needs [29]. Moreover, poor sleep quality is more prevalent among female athletes and those in aesthetic sports [29].

Athletes tend to have similar, or slightly better, sleep duration and quality compared to sedentary individuals, with moderate exercise being recommended for improved sleep. However, many athletes average less than 8 h of sleep across various sports, with no significant difference between genders [30]. Elite athletes often report poor sleep quality, particularly before competitions due to mood and anxiety disturbances.

Sleep deprivation negatively affects athletic performance and recovery [25,31]. Several studies have connected sleep deprivation with poor performance (including cardiorespiratory and psychomotor consequences), mental health, injury recovery, and is an injury risk factor [25,31,32]. Lack of sleep negatively impacts physiological and psychological performance, primarily affecting mood, decision-making skills, and cognitive functioning. Insufficient sleep can impair one’s decision-making processes and reduce performance outcomes in sports. Physiological consequences of sleep loss include weakened immune functioning, reduced submaximal sustained performance, and impaired glucose metabolism, potentially leading to increased levels of fatigue [33,34,35,36,37,38].

Several questionnaires may be utilized to evaluate sleep quality and hygiene. Due to their simple administration and affordability, questionnaires are a valuable tool for preliminary sleep evaluations and screenings. Widely utilized questionnaires in insomnia research encompass the Pittsburgh Sleep Quality Index (PSQI) for gauging sleep quality, the Sleep Hygiene Index for examining sleep hygiene, and the Epworth Sleepiness Scale for measuring daytime drowsiness [39,40,41]. Nevertheless, these questionnaires lack validation regarding athletes (see Table 1). Athlete-centric questionnaires comprise the Athlete Sleep Screening Questionnaire and the Athlete Sleep Behavior Questionnaire, which examine sleep hygiene. However, it is not a sleep quality screening tool [41]. Furthermore, owing to a shortage of translated questionnaires, few studies in the Arabian community assess sleep quality.

Consequently, it is critical to develop a valid and accurate questionnaire that can be used for screening and identifying athletes with clinically significant sleep issues and disorders. This screening would allow for prompt action when necessary and identify individuals who may require education and behavioral intervention.

The Athlete Sleep Screening Questionnaire (ASSQ) was developed as a screening tool to identify clinically significant sleep disturbances and daytime dysfunctions among athletes and to prescribe interventions based on the nature and severity of the discovered sleep disturbances [39]. It is a 15-item questionnaire that investigates sleep and circadian features such as sleep quantity, sleep quality, insomnia, and chronotype [16]. The ASSQ accurately identifies athletes that would benefit from preventive measures and those that suffer from clinically significant sleep problems requiring a specialist physician [42]. Additionally, the ASSQ is recommended by the International Olympic Committee for sleep quality screening [43]. Notably, ethnicity may influence sleep characteristics [44].

Given the importance of sleep and increased sports participation among Arabian athletes, translating and validating the ASSQ is crucial. Therefore, this study aimed to translate the ASSQ into Arabic and test its convergent validity and reliability among Arabic-speaking athletes. Moreover, this study hypothesized that the Arabic-translated ASSQ would be valid and reliable to use in clinical practice.

## 3. Materials and Methods

### 3.1. Study Design

Several steps were followed to translate and validate the ASSQ for Arabic-speaking athletes. The translation process followed Beaton’s guidelines [45]. The flow chart illustrating study participation is presented in Figure 1. Additionally, informed consent was obtained from all participants, and the study was approved by Prince Sattam Bin Abdulaziz ethical committee No.: RHPT/021/016.

### 3.2. Translation Procedure

The original developer of the ASSQ approved its translation [39]. The translation and cultural adaptation adhered to Beaton et al.’s [45] cross-cultural adaptation guidelines. This comprehensive process entailed six steps as follows:

Step I: Two professional bilingual translators with Arabic as their native language independently translated the English ASSQ into Arabic, resulting in two distinct Arabic translations.

Step II: A unified translation was created by two bilingual Arabic/English speakers who compared and combined the two initial translations.

Step III: Two new independent professional bilingual translators, native English speakers unfamiliar with the original ASSQ, independently performed the backward translation of the ASSQ from Arabic to English. The resulting translations were submitted to a committee (Step IV).

Step IV: Four researchers, including all Step I and II translators, formed an expert committee that examined every translation report. They aimed to ensure equivalency between the original ASSQ and the Arabic-adapted version (ASSQ-Ar). Moreover, 12 participants were required to respond to each ASSQ test question as part of the pre-testing process.

### 3.3. Participants

There is no consensus regarding the method for calculating sample size for translation studies or questionnaire validation. Recommendations for sample size in factor analysis range from three to 10 individuals per variable (item), with a minimum of 50 participants. [46,47,48] Consequently, the present study requires at least 80 participants. Through convenience sampling, 90 athletes were recruited to complete the Arabic ASSQ and PSQI from June to December 2021. The athletes were recruited from three-sport clubs and two sport rehabilitation clinics. The inclusion criteria included: (1) 18–45 years of age, (2) ability to understand the written Arabic language, (3) practice sports regularly, and (4) consent to participate in the study. The exclusion criteria included a rheumatic or neurological disease.

### 3.4. Procedure

Two trained physical therapists recruited and examined the athletes during the pre- and final testing of the ASSQ. The eligible athletes completed the ASSQ and PSQI, and their characteristics, such as age, sex, height, and weight, were collected. This measurement was repeated within one week after completing the baseline outcome measures. During the ASSQ-Ar pre-testing stage, clarity, comprehension, and baseline and follow-up time were assessed. Participants were able to comprehend and complete the questionnaires within 10 min.

### 3.5. Outcome Measures

The ASSQ is valid and reliable in assessing athletes’ clinically significant sleep problems. Eleven questions from the ASSQ were used to determine whether an athlete required further assessment and treatment specific to sleep-related problems [43]. Of these, five questions (1, 3, 4, 5, and 6) were used to calculate the ASSQ-Sleep Difficulty Score (ASSQ-SDS) on an 18-point scale. The remaining questions were not included in the final score. However, these characteristics guided sleep improvement techniques, such as increasing napping frequency, reducing caffeine intake, and limiting electronic device use before sleeping. A higher ASSQ-SDS score indicated poor sleep. ASSQ-SDS scores can be used to classify athletes into four groups of clinical sleep problems: none (0–4), mild (5–7), moderate (8–10), and severe (11–17). Questions 7–10 of the ASSQ were used to score the chronotype, the inherent tendency of one’s body to sleep at a given period, on a 15-point scale to assess sleep difficulty. A total score of fewer than five points on the ASSQ-chronotype implied that an athlete had an evening chronotype.

The PSQI [49] is a validated 19-item scale that differentiates between poor and good sleepers. The overall PSQI score ranges from 0–21 and comprises seven items: sleep quality, latency, duration, efficiency, disturbances, use of sleep medication, and daytime dysfunction. The cut-off score for poor sleepers is >5, with sensitivity (89.6%) and specificity (86.5%).

### 3.6. Statistical Analysis

The score distributions of the ASSQ-SDS and ASSQ-chronotypes were analyzed to determine floor and ceiling effects. These effects corresponded to the proportion of participants with the lowest or highest scores on the ASSQ-SDS and ASSQ-chronotype. To rule out any compromise in the reliability and validity of the scales, the proportion of participants with the lowest and highest scores should not exceed 15% [49]. Cronbach’s alpha was used to estimate the internal consistency. Cronbach’s alpha between 0.70 and 0.90 implies excellent internal consistency [50]. Test-retest reliability was calculated using the intraclass correlation coefficient (ICC) between the first and second administrations (two-way random-effects model, single measure) and its 95% confidence interval (CI). ICC values exceeding 0.75 indicate excellent reproducibility [51]. The standard error of measurement (SEM) was calculated using the formula: SEM = SD_pooled standard deviation_/√2 [26]. The smallest detectable change in the individual score (SDC_individual_) was calculated using the following formula: SDC_individual_ = 1.96 × √2 × SEM. The smallest detectable change for the group score (SDC _group_) was further calculated using the following formula: SDC_group_ = SDC_individual_/√n. Wilcoxon tests compared the first and second administrations of the ASSQ-SDS and ASSQ-chronotypes. Bland-Altman plots (including mean difference ± 1.96 × SD of the difference) were constructed for the ASSQ-SDS and ASSQ-chronotypes to examine the agreement and check for systematic bias.

Using Spearman’s rank correlation coefficient, Convergent validity was determined to compare the ASSQ and PSQI results. Correlations of 0.90–1.00, 0.70–0.90, 0.50–0.70, 0.30–0.50, and 0.00–0.30 were considered extremely high, high, moderate, low, and insignificant, respectively. Statistical significance was set at *p* < 0.05. Data were analyzed using the statistical package for social sciences (IBM SPSS, Chicago, IL, USA, Version 23).

## 4. Results

### 4.1. Cross-Cultural Adaptation

The final ASSQ-Ar (Appendix A) was tested among 12 athletes (seven men and five women) to evaluate the comprehensibility of the final version. After completing the ASSQ-Ar, the participants were asked if they had any questions or thoughts to ensure that they understood the questionnaire. The research team made notes and evaluated all reports. ASSQ-Ar questions were found to be clear and easy to understand. Moreover, the questionnaire was found to be culturally appropriate without any issues or discrepancies. Consequently, the final ASSQ-Ar sample was not altered.

### 4.2. Study Participants

A total of 90 athletes (mean age 25.72 ± 4.99 years) met the inclusion criteria and completed the questionnaires. Of these, 26 were female (28.9%), and 64 were male (71.1%); 42 participants (46.7%) reported playing soccer as their favorite sport, followed by CrossFit (*n* = 8, 8.9%) and boxing (*n* = 5, 5.6%). The remaining participants reported preferring different sports.

### 4.3. Floor and Ceiling Effects

The average total score on the ASSQ-SDS was 6.49 ± 2.69 (range = 1–12). The average total score on the ASSQ-chronotype was 7.52 ± 2.97 (range = 1–14). The ASSQ-SDS and ASSQ-chronotype had no floor or ceiling effects. Regarding the floor effect, none of the participants scored less than one on both the ASSQ-SDS and ASSQ-chronotype. Regarding the ceiling effect, none of the participants scored more than 17 and 14 on the ASSQ-SDS and ASSQ-chronotypes, respectively.

### 4.4. Internal Consistency

The internal consistency of the ASSQ-SDS, based on the correlation strength among five questions (1, 3, 4, 5, and 6), was poor, with a Cronbach’s alpha of 0.435. Similarly, the internal consistency of the ASSQ-chronotype, based on the correlation strength among the remaining four questions (7–10), was also poor, with a Cronbach’s alpha of 0.632.

### 4.5. Test-Retest Reliability

Test-retest reliability was assessed among 81 athletes who completed the questionnaire twice, averaging 5 ± 2 days between tests. The test-retest reliability of the ASSQ-SDS and ASSQ-chronotype was excellent, with an ICC of 0.84 and 0.938, respectively (Table 2). Regarding the ASSQ-SDS, the SEM was 1.91, SDC_individual_ was 5.30, and SDC_group_ was 0.59. Regarding the ASSQ-chronotype, the SEM was 2.11, SDC_individual_ was 5.86, and SDC_group_ was 0.65. In addition, the Wilcoxon test showed that the first and second administration scores on the ASSQ-SDS differed significantly (*p* = 0.043). In contrast, the difference between the first and second administration scores on the ASSQ-chronotype was not statistically significant (*p* = 0.295). The Bland-Altman plots of the ASSQ-SDS (Figure 2) and ASSQ-Chronotype (Figure 3) showed no systematic bias.

### 4.6. Convergent Validity

The results showed that the ASSQ-SDS score was significantly and positively correlated with the PSQI (Spearman’s rho = 0.734, *p* < 0.001). Additionally, the ASSQ-chronotype was significantly negatively correlated with the PSQI (Spearman’s rho = −0.549, *p* < 0.001).

## 5. Discussion

This study aimed to cross-culturally adapt, translate, and validate the English version of the ASSQ into Arabic. The results show that the Arabic version of the ASSQ has satisfactory psychometric properties similar to the original English version.

Moreover, the ASSQ-Ar has excellent test-retest reliability, similar to the original ASSQ development study and the Chinese version [51,52]. Additionally, a moderate to high correlation was found between the Arabic versions of the ASSQ and PSQI for sleep screening among Saudi athletes, consistent with previous findings [39]. These results indicate the applicability of the ASSQ-Ar for daily sleep monitoring and screening among Saudi athletes.

The test-retest reliability of the Arabic version of the ASSQ-SDS and ASSQ-chronotype was similar to a recently translated Chinese version of the ASSQ. However, the Chinese version only reported the overall test-retest reliability and did not consider the scale subcategories [53]. In addition, the present study’s findings were comparable to the original development of the scale, where test-retest reliability was reported as good [40].

Both subscales of the ASSQ (SDS and chronotype) yielded poor internal consistency (0.435 and 0.632, respectively). This indicated that some items of the ASSQ-SDS and ASSQ-chronotype subscales only correlated weakly. This outcome is attributable to the small sample size and its heterogeneity regarding the broad age range, different sports played, and various cultural backgrounds. Some studies proposed that females experience significantly better sleep quality than males within the same age group [39]. However, other researchers argued that sleep deprivation effects are similar for both sexes. Additionally, a combination of factors such as the frequency of training sessions, psychological stress associated with pre-competition preparation, and other external influences can also alter athletes’ sleep patterns [39].

Variability in sleep patterns can also be attributed to differences in sport competition types and training phases. This finding also indicates that some items should be adjusted or eliminated in future studies to improve the reliability and internal consistency of the subscales. Another possible explanation for these findings may be that Cronbach’s alpha is dependent on the number of items within the scale, suggesting that a low number of items within the scale can lead to a low Cronbach’s alpha [53]. The ASSQ-SDS and ASSQ-chronotype only have a small number of items, which may be the reason for the low internal consistency. Moreover, it should be noted that in the original ASSQ study, Cronbach’s alpha was only slightly above 0.7 (the acceptable threshold) for both subscales: 0.74 for ASSQ-SDS and 0.73 for ASSQ-chronotype.

The current study found a moderate correlation between PSQI and ASSQ-SDS and a high correlation between PSQI and ASSQ-chronotype. Additionally, Tan et al. [54] reported moderate convergent validity between the overall score of the Chinese version of ASSQ and PSQI. These findings indicate that the ASSQ-Ar may be a good tool for screening and evaluating sleep among athletes.

Clinical implications for screening for sleep problems are essential, particularly for athletes and teams with restricted time during the season. Objective measures of sleep, including polysomnography, actigraphy, and interviews, are clinically valid, high-quality diagnostic tools. However, administering them is time-consuming and resource intensive. In contrast, questionnaires are clinically valid and extensively used to screen for sleep issues. The ASSQ is time-efficient and cost-effective for screening clinically relevant sleep issues among athletes [16,19,30]. In addition, this questionnaire is used to determine whether an athlete requires further sleep-related assessment and intervention. Previous research clinically validated the ASSQ using a sleep medicine physician [42]. This study found that the ASSQ is 81% sensitive and 93% specific in identifying athletes with clinical sleep disturbances who require further assessment from a sleep specialist. Therefore, the ASSQ is easy to use, quick to complete, and remotely scored if necessary [42]. Future research should examine the ASSQ with objective sleep measures, including polysomnography and actigraphy.

This study is the first to translate and validate a scale that screens and assesses sleep among Saudi athletes. However, it has several limitations that should be addressed in future research. The ASSQ-Ar has excellent sensitivity and specificity and significantly correlates with subjective self-reported measures of the PSQI. However, it requires further evaluation to understand the validation and discrepancy between objective and subjective sleep measures. This study includes a wide age range (18–45 years) with a small sample size to indicate sleep quality changes with age. Including a large sample size will improve the generalizability of this study to Arabic athletes. Another potential limitation is that prescribed medication for sleep health may have side effects. However, the PSQI has one component that considers medication a risk factor for poor sleep quality. The data used in this study were collected from healthy athletes; therefore, the results do not apply to patients receiving sleep treatment. Including a control group and a longitudinal design will contribute to understanding the consistent validation and causal conclusions.

## 6. Conclusions

Overall, this study provides valuable insights for clinicians to evaluate sleep disturbance among athletes easily. The ASSQ-Ar has good psychometric properties, is quick to administer, easy to complete, and has the advantage of remote scoring. However, future studies should investigate the significance of using ASSQ to detect sleep improvement after participating in sleep promotion programs.

## Figures and Tables

**Figure 1 healthcare-11-01501-f001:**
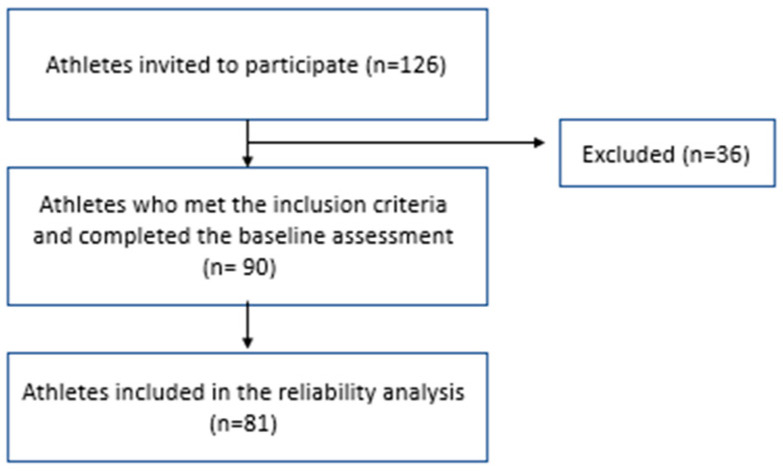
Recruitment process.

**Figure 2 healthcare-11-01501-f002:**
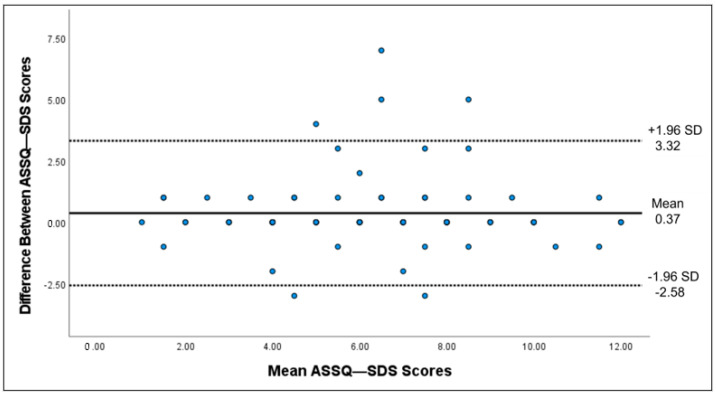
Bland-Altman plot of the ASSQ-SDS, including the mean difference and the limits of agreement.

**Figure 3 healthcare-11-01501-f003:**
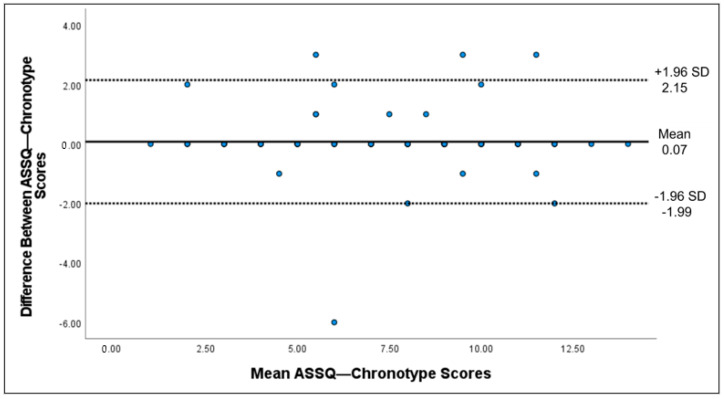
Bland-Altman plot of the ASSQ-chronotype, including the mean difference and the limits of agreement.

**Table 1 healthcare-11-01501-t001:** Demographics characteristics of participants.

Gender	
Male (n = 64)	28.9%
Female (n = 26)	71.1%
Hight (mean ± SD)	170.08 ± 7.7
Wight (mean ± SD)	71.01 ± 9.01
BMI (mean ± SD)	22.03 ± 3.30
Smokers, yes (n, %)	10, (10.09%)

**Table 2 healthcare-11-01501-t002:** The test-retest reliability of the ASSQ-SDS and ASSQ-Chronotype (n = 81).

	ASSQ-SDS	ASSQ-Chronotype
First test	6.49 ± 2.72	7.63 ± 2.98
Second test	6.12 ± 2.69	7.56 ± 3.00
Mean difference	0.37	0.07
ICC (95% CI)	0.84 (0.75–0.89)	0.93 (0.90–0.96)
SEM	1.91	2.11
SDC_individual_	5.30	5.86
SDC_group_	0.59	0.65

Data are presented as mean ± standard deviation. ASSQ: Athlete Sleep Screening Questionnaire; SDS: sleep difficulty score; ICC: intraclass correlation coefficient; CI: confidence interval; SEM: standard error of measurement; SDC: smallest detectable change.

## Data Availability

All data are available from the principal investigator with a reasonable request.

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
