# Peer review of "Translation and Validation of the Arabic Version of the Athlete Sleep Screening Questionnaire"

_healthcare, 2023, doi:10.3390/healthcare11101501_

Round 1

Reviewer 1 Report

I appreciate the opportunity to review this manuscript, which is interesting and relevant to the quality of life of high performance athletes's study. I congratulate the authors for the work done, however, there are reasons why I consider that it is not publishable, the main one being poor reliability, and in general the following aspects:

Introduction

The authors write that sleep difficulties occur more in high performance sports, also in the sample they say that they were elite athletes; however, it is questionable whether crossFit practitioners are high performers. Apparently the sample are not elite athletes.

It would be convenient to define chronotype to better understand the function of the instrument.

Clearly write the objective, indicating the type of validity that is tested.

Method

The authors mention that the ASSQ has 15 items, while the results read only 5 items for ASSQ-SDS and 4 items for ASSQ-chronotype. It is not clear what happened to the rest of the items, as well as the way to respond.

They mention that the construct validity is tested. It would be clearer if it is pointed out that convergent and discriminant validity were analyzed.

In the procedure, it is not clear what the pre-testing consisted of. It is also unclear who the athletes eligible for the second application were.

Results

Reliability is poor, therefore, it does not ensure that the items are understood correctly, or that they accurately measure the associated constructs. Review the item response theory for a better analysis.

Attorresi, H. F., Lozzia, G. S., Abal, F. J. P., Galibert, M. S., & Aguerri, M. E. (2009). Item Response Theory. Basic concepts and applications for the measurement of psychological constructs. Argentine Journal of Clinical Psychology, 18(2), 179-188.

The test-retest confirms that the items are imprecise both in the first and in the second application.

Regarding the Wilcoxon test, they were asked if it is normal for there to be differences in ASSQ-SDS between the second and first application, being the same participants without an intervention in sleep management. If the authors expect differences, the reason should be explained.

Discussion

The authors mention that the results may be attributable to cultural contexts, the question is which contexts.

The authors say that the items should be adjusted or removed. Why didn't you do it? That would be a contribution to the study. They are asked to review the item response theory to try to improve reliability.

The authors say “The ASSQ is effective in terms of time and cost in detecting clinically relevant sleep-related problems among athletes” however, the instrument is not reliable or accurate, therefore its use, and the results it offers, are questionable. to evaluate sleep disturbance among athletes.

The conclusion seems invalid.

Author Response

Thank you for allowing us to submit a revised draft of our manuscript titled “Translation and Validation of the Arabic Version of the Athlete Sleep Screening Questionnaire.” We appreciate the time and effort to providing your valuable feedback on our manuscript. We are grateful to for your insightful comments on our paper. We have been able to incorporate changes to reflect most of the suggestions . 

Here is a point-by-point response to  comments and concerns.

Introduction

  1. The authors write that sleep difficulties occur more in high performance sports, also in the sample they say that they were elite athletes; however, it is questionable whether crossFit practitioners are high performers. Apparently the sample are not elite athletes.

Response: Thank you for your feedback. The sample comprised athletes from different sports. However, all of them were or used to be elite sports players.

  1. It would be convenient to define chronotype to better understand the function of the instrument.

Response: Thank you for your feedback. The definition of chronotype was added in the method section.

  1. Clearly write the objective, indicating the type of validity that is tested.

Response: Thank you for your comment. The type of validity is now included within the objectives of the study.

Method

The authors mention that the ASSQ has 15 items, while the results read only 5 items for ASSQ-SDS and 4 items for ASSQ-chronotype. It is not clear what happened to the rest of the items, as well as the way to respond.

Response: Thank you for your comment. The remaining items were not included in the scoring as they focus on informing sleep optimization strategies, such as strategies to increase napping frequency, reduce caffeine intake, and reduce electronic device use before bedtime.

  1. They mention that the construct validity is tested. It would be clearer if it is pointed out that convergent and discriminant validity were analyzed.

Response: We used convergent validity. This has been clarified in the revised manuscript (in the abstract and in the results section).

  1. In the procedure, it is not clear what the pre-testing consisted of. It is also unclear who the athletes eligible for the second application were.

Response: Thank you for your feedback. Further explanations of the pre-testing stage were included in the procedures section.

Results

  1. Reliability is poor, therefore, it does not ensure that the items are understood correctly, or that they accurately measure the associated constructs. Review the item response theory for a better analysis.
    Attorresi, H. F., Lozzia, G. S., Abal, F. J. P., Galibert, M. S., & Aguerri, M. E. (2009). Item Response Theory. Basic concepts and applications for the measurement of psychological constructs. Argentine Journal of Clinical Psychology, 18(2), 179-188.

Response: We acknowledge the reviewer’s concern regarding the poor internal consistency for both subscales. We appreciate the reference provided by the reviewer, however, we were unable to find the English version of the study (we only found the Spanish version available at (https://www.proquest.com/openview/55abffd9306bf48a3ff163a96c0cd588/1?pq-origsite=gscholar&cbl=4380457).

Besides the reasons discussed in the manuscript , another possible explanation for our findings could be that the Cronbach’s alpha is dependent on the number of items within the scale, as a low number of items within the scale can lead to a low Cronbach’s alpha. The ASSQ-SDS and ASSQ-chronotype in our study only have a small number of items, which may be the reason for the low internal consistency. Moreover, we note that in the original study of the ASSQ, the Cronbach’s alpha was only slightly above 0.7 (the acceptable threshold) for both subscales: 0.74 for ASSQ-SDS and 0.73 for ASSQ-chronotype.

  1. The test-retest confirms that the items are imprecise both in the first and in the second application.

Response: We appreciate the reviewer’s comment. Our results indicate that the test-retest reliability of both the ASSQ-SDS and the ASSQ-chronotype was excellent, with ICCs of 0.84 and 0.938, respectively. As such, we are uncertain as to how the results of test-retest reliability “confirms that the items are imprecise both in the first and in the second application.” We would greatly appreciate it if the reviewer could provide further clarification on this point.

  1. Regarding the Wilcoxon test, they were asked if it is normal for there to be differences in ASSQ-SDS between the second and first application, being the same participants without an intervention in sleep management. If the authors expect differences, the reason should be explained.

Response: We appreciate your question regarding this matter. We did not anticipate a significant difference between the mean scores of the first and second administrations of the ASSQ-SDS and ASSQ-chronotype, given that the average time between tests was 5 ± 2 days. To further investigate this issue, we employed the Wilcoxon test for statistical analysis. Contrary to our expectation, we found a significant difference between the first and second administrations of the ASSQ-SDS (albeit the p-value was 0.043), but not for the ASSQ-chronotype.

Discussion

  1. The authors mention that the results may be attributable to cultural contexts, the question is which contexts.

Response: Thank you for your feedback. Cultural aspects differ between populations, including Saudi Arabia, which affects sleep and lifestyle. Sleep pattern is commonly affected in Saudi Arabia owing to social gathering, coffee consumption, tea consumption, delaying sleep time as well as shorter sleep duration. We added this to the paragraph in the discussion section.

  1. The authors say that the items should be adjusted or removed. Why didn't you do it? That would be a contribution to the study. They are asked to review the item response theory to try to improve reliability.

Response: Thank you for your valuable feedback. This is our future research plan as this requires a larger sample size and Rasch analysis that cannot be done with small samples. We already mentioned that future research should examine this and we included this in the limitation section.

  1. The authors say “The ASSQ is effective in terms of time and cost in detecting clinically relevant sleep-related problems among athletes” however, the instrument is not reliable or accurate, therefore its use, and the results it offers, are questionable. to evaluate sleep disturbance among athletes.

Response: Thank you for your feedback. In fact, the instrument had excellent reliability. We explained this in the results section “The test-retest reliability of the ASSQ-SDS and ASSQ-chronotype was excellent, with an ICC of 0.84 and 0.938, respectively”. Additionally, convergent validity was good with another scale “PSQI”. We also explained this in the results “The results showed that the ASSQ-SDS score was significantly and positively correlated with the PSQI (Spearman’s rho = 0.734, P < 0.001). Additionally, the ASSQ-chronotype was significantly negatively correlated with the PSQI (Spearman’s rho = – 0.549, P < 0.001).” Only the subscales had poor internal consistency and we explained the possible reasons for these results in the discussion section. 

  1. The conclusion seems invalid.(Ahmed)

Response: We revised this and removed identifying clinically significant sleep disorders.

Reviewer 2 Report

1.     One of the contributions claimed by the authors is that the proposed new method. This is incorrect. Simply converting an old model is not the definition of new model. And the model description does not support this claim.

2.     The data description is confusing. And  is inconsistent with the tables.

3.     Figure 1 can be better used for illustration by adding the description of each term A part of the current introduction section contains the existing studies in the field. It is recommended to put them into a separate related work section.

4.     There are grammar/expression issues throughout the paper. A careful round of proofreading is needed.

Author Response

Thank you for allowing us to submit a revised draft of our manuscript titled “Translation and Validation of the Arabic Version of the Athlete Sleep Screening Questionnaire.” We appreciate the time and effort that you and the reviewers have dedicated to providing your valuable feedback on our manuscript. We are grateful to the reviewers for their insightful comments on our paper. We have been able to incorporate changes to reflect most of the suggestions provided by the reviewers.

Here is a point-by-point response to the comments and concerns.

  1. One of the contributions claimed by the authors is that the proposed new method. This is incorrect. Simply converting an old model is not the definition of new model. And the model description does not support this claim.

Response: Thank you for your response. We have edited the manuscript accordingly.

  1. The data description is confusing. And is inconsistent with the tables.

Response: Thank you for your response. We have edited the manuscript accordingly.

  1. Figure 1 can be better used for illustration by adding the description of each term A part of the current introduction section contains the existing studies in the field. It is recommended to put them into a separate related work section.

Response: The submitted manuscript contained no figures.

Reviewer 3 Report

It is nice study that conduct a validation study on sleep quality screening in athletic population and arabic language. I have add my minor suggestion to pdf version of your paper. 

Author Response

Dear Editor:

Thank you for allowing us to submit a revised draft of our manuscript titled “Translation and Validation of the Arabic Version of the Athlete Sleep Screening Questionnaire.” We appreciate the time and effort that you and the reviewers have dedicated to providing your valuable feedback on our manuscript. We are grateful to your insightful comments on our paper.

On the revised manuscript, we were able to incorporate changes to reflect the suggestion. 

Reviewer 4 Report

1- General comment

Dear authors,

I congratulate your effort and time spent conducting this study. The aim was to analyze the validity and reliability of the Arabic version of the Athlete Sleep Screening Questionnaire (ASSQ-AR). In general, the results showed that the ASSQ-AR presented satisfactory psychometric properties similar to the original English version, which can provide valuable insights for clinicians to evaluate sleep disturbance among athletes. Although the manuscript has the potential for publication due to the relevant topic, there is space for improvement, namely in the introduction and methods section. Below, I provide some comments and suggestions.

2- Abstract

This section should be posteriorly reformulated based on the commentaries and suggestions below.

3- Introduction

LL36-39: The paragraph would benefit from more examples of the benefits of sleeping well for athletic development and the disadvantages of sleep deprivation.

LL40-43: The authors should provide a more comprehensive perspective on the different tools to analyze sleep quality. For example, start by mentioning the gold standard and expensive tools and then the importance of using simple, easy-to-use, and quick-to-administer questionnaires.

LL44: “Several outcome measures…”. The authors should replace the term “outcome measures” with a more appropriate one. Maybe “questionnaires”.

LL57-58: The authors should present the research problem or gap between the last and penultimate paragraphs. What research problem needs to be explored in the current study? This information should be explicit for the readers.

LL61: The authors should present the study hypothesis based on previous and similar research.

4- Materials and methods

The authors should start this section with the “2.1 Study design” subsection, where they exactly describe the experimental procedures. Currently, I think the procedures performed to run the study are poorly understood. For example, what did they do on days 1 and 2? This information should be explicit for the reader. Please support this information by illustrating the procedures through a figure.

LL63-65: “The ASSQ was translated into Arabic and tested for floor or ceiling effects, internal consistency, reproducibility, and validity among Arabic-speaking athletes. In addition, the PSQI was used to evaluate construct validity.”. The authors should remove these sentences, as they reflect the study’s aims.

LL65-67: This information should be placed in the participant’s subsection.

LL81-82: Please, replace with the following: “Ninety athletes were recruited to complete the Arabic ASSQ and PSQI from June to December 2021.”

LL82: The inclusion criteria do not mention elite athletes, which makes me wonder if the authors included 90 elite athletes. There is no information in the entire manuscript regarding the characteristics of the included elite athletes. What are the type of sports, training experience, training habits, clubs/teams? I just read in lines 143-144 that “Forty-two participants (46.7%) reported playing soccer as their favorite sport, followed by CrossFit (n = 8, 8.9%) and boxing (n = 5, 5.6%).”. Nevertheless, I have to say that an elite athlete does not report his/her favorite sport because an elite athlete competes in a specific sport. Therefore, based on my concerns, I strongly suggest the authors remove the word “elite” from the manuscript and indicate that the included participants practice sports regularly (this should be written in the inclusion criteria).

LL88-89: Where is the information regarding age, height, and weight?

LL92-93: Please, support the sentence with appropriate references.

LL94-96: Why the authors only used five questions (1, 3, 4, 5, and 6)? What were the criteria? This decision should be clarified.

LL102-103: Please, support the sentence with appropriate references.

LL107: The statistics chosen to determine the validity and reliability seem correct. Nevertheless, the variables you used to analyze the reliability are poorly perceived. Are you analyzing the inter-reliability (i.e., the comparison of the results between two different questionnaires) or intra-reliability (i.e., the comparison of the results of the same questionnaire)? For example, you compared the first and second measurements of the ASSQ-SDS and ASSQ-Chronotype for the ICC analysis. However, this is only describing the intra-reliability. I think the authors should also analyze the inter-reliability (ASSQ-SDS and ASSQ-Chronotype vs. PSQI).

Besides these changes, the authors should also provide more information to improve the robustness of the analysis. First, the authors should present the required sample size for the selected reliability coefficients. You can determine the sample size based on one reliability coefficient (e.g., Cronbach’s alpha or ICC). For example, if you use Cronbach’s alpha, you must indicate the minimum acceptable Cronbach’s alpha, expected Cronbach’s alpha, significance level, statistical power, number of items, and expected dropout rate. On the other hand, if you use ICC, you must indicate the minimum acceptable ICC, expected ICC, significance level, statistical power, number of raters or repetitions per subject, and expected dropout rate. Please, follow this article to determine the sample size: https://doi.org/10.21315/eimj2018.10.3.8.

Second, Bland-Altman plots with 95% limits of agreement (LOA) should be presented to demonstrate the systematic bias/differences between results. You can follow the article “Measuring agreement in method comparison studies” by Bland and Altman (DOI: 10.1177/096228029900800204).

LL119-121: For SEM, you should first calculate the standard deviation of the differences between scores and then divide by √2 (see this book https://doi.org/10.1017/CBO9780511996214 to understand the calculations and why the authors discourage using the formula SEM = SD x √(1-ICC)). To calculate SDC, multiply the SEM by 1.96 and √2, as you already did. I also suggest presenting SEM and SDC’s relative (%) values.

5- Results

LL141: Please, complement the text with a Table ( characteristics of the participants). More information should be added, including sex, age, body height, body mass, and sport practiced.

After “3.4. Internal consistency”, please report the systematic bias between the first and second administrations for the ASSQ-SDS and ASSQ-chronotypes, and the differences between ASSQ-SDS vs. PSQI and ASSQ-Chronotype vs. PSQI.

LL160: The authors should also report the inter-reliability analysis (ASSQ-SDS vs. PSQI and ASSQ-Chronotype vs. PSQI).

6- Discussion

I suggest the authors structure the discussion section with the following subsections:

i) “Main findings”: succinctly describe the results;

ii) “Internal consistency and systematic bias”: discuss and compare the results (Cronbach’s alpha and Bland-Altman plots) with the available literature;

iii) “Test-retest reliability”: discuss and compare the results (intra and inter-reliability) with the available literature;

iv) “Construct validity”: discuss and compare the results (correlations) with the available literature;

v) “Clinical implications”: describe the clinical and sports implications;

vi) “Strengths and limitations of the study”: present the main strengths and limitations of the study with a focus on future research.

Author Response

Thank you for allowing us to submit a revised draft of our manuscript titled “Translation and Validation of the Arabic Version of the Athlete Sleep Screening Questionnaire.” We appreciate the time and effort that you and the reviewers have dedicated to providing your valuable feedback on our manuscript. We are grateful to the reviewers for their insightful comments on our paper. We have been able to incorporate changes to reflect most of the suggestions provided by the reviewers. We have highlighted the changes within the manuscript.

Here is a point-by-point response to the reviewers’ comments and concerns.

Abstract (Ahmed)

This section should be posteriorly reformulated based on the commentaries and suggestions below.

Response: Thank you for your response. We have edited the abstract accordingly.

3- Introduction (Ahmed)

LL36-39: The paragraph would benefit from more examples of the benefits of sleeping well for athletic development and the disadvantages of sleep deprivation.

Response: Thank you for your comment. The paragraph was edited to include examples of the sleep deprivation effect.

LL40-43: The authors should provide a more comprehensive perspective on the different tools to analyze sleep quality. For example, start by mentioning the gold standard and expensive tools and then the importance of using simple, easy-to-use, and quick-to-administer questionnaires.

 Response: Thank you for your comment. We have moved the paragraph that discusses the gold standard tools and edited the paragraph to capture the intended comment.

LL44: “Several outcome measures…”. The authors should replace the term “outcome measures” with a more appropriate one. Maybe “questionnaires”.

 Response: Thank you for your comment. We have edited the term.

LL57-58: The authors should present the research problem or gap between the last and penultimate paragraphs. What research problem needs to be explored in the current study? This information should be explicit for the readers.

  Response: Thank you for your comment. We have edited the paragraph to explore the need for the study.

LL61: The authors should present the study hypothesis based on previous and similar research.

  Response: Thank you for your comment. We have included an explanation on the hypothesis generation.

4- Materials and methods (Faris)

The authors should start this section with the “2.1 Study design” subsection, where they exactly describe the experimental procedures. Currently, I think the procedures performed to run the study are poorly understood. For example, what did they do on days 1 and 2? This information should be explicit for the reader. Please support this information by illustrating the procedures through a figure.

  Response: Thank you for your comment. We have added a paragraph to illustrate the procedure.

LL63-65: “The ASSQ was translated into Arabic and tested for floor or ceiling effects, internal consistency, reproducibility, and validity among Arabic-speaking athletes. In addition, the PSQI was used to evaluate construct validity.”. The authors should remove these sentences, as they reflect the study’s aims.

 Response: Thank you for your response. We have edited the paragraph accordingly.

LL65-67: This information should be placed in the participant’s subsection.

 Response: Thank you for your response. We have edited the paragraph accordingly.

LL81-82: Please, replace with the following: “Ninety athletes were recruited to complete the Arabic ASSQ and PSQI from June to December 2021.”

 Response: Thank you for your response. We have edited the paragraph accordingly.

LL82: The inclusion criteria do not mention elite athletes, which makes me wonder if the authors included 90 elite athletes. There is no information in the entire manuscript regarding the characteristics of the included elite athletes. What are the type of sports, training experience, training habits, clubs/teams? I just read in lines 143-144 that “Forty-two participants (46.7%) reported playing soccer as their favorite sport, followed by CrossFit (n = 8, 8.9%) and boxing (n = 5, 5.6%).”. Nevertheless, I have to say that an elite athlete does not report his/her favorite sport because an elite athlete competes in a specific sport. Therefore, based on my concerns, I strongly suggest the authors remove the word “elite” from the manuscript and indicate that the included participants practice sports regularly (this should be written in the inclusion criteria).

  Response: Thank you for your response. We have removed the word elite and provided a table of sample characteristics.

LL88-89: Where is the information regarding age, height, and weight?

Response: The information regarding age has been added to the revised manuscript  

LL92-93: Please, support the sentence with appropriate references.

 Response: Thank you for your response. We have provided the reference.

LL94-96: Why the authors only used five questions (1, 3, 4, 5, and 6)? What were the criteria? This decision should be clarified.

Response: The original authors of the questionnaire removed two questions from the ASSQ-SDS (which originally comprised seven questions and was reduced to five questions) to improve the internal consistency from being poor (Cronbach’s alpha = 0.58; 95% CI 0.50 to 0.66) to acceptable (Cronbach’s alpha = 0.74; 95% CI 0.69 to 0.79). These two questions were related to “how often the athlete napped” and “the duration of the nap,” which correlated poorly with the SDS total score (r = 0.16 and r = 0.04, respectively). According to the original authors, the question regarding the duration of the nap was dropped from the questionnaire; however, the question regarding the nap frequency (question #2) was kept to provide information related to sleep education strategies.

LL102-103: Please, support the sentence with appropriate references.

  Response: Thank you for your response. We have provided the reference.

LL107: The statistics chosen to determine the validity and reliability seem correct. Nevertheless, the variables you used to analyze the reliability are poorly perceived. Are you analyzing the inter-reliability (i.e., the comparison of the results between two different questionnaires) or intra-reliability (i.e., the comparison of the results of the same questionnaire)? For example, you compared the first and second measurements of the ASSQ-SDS and ASSQ-Chronotype for the ICC analysis. However, this is only describing the intra-reliability. I think the authors should also analyze the inter-reliability (ASSQ-SDS and ASSQ-Chronotype vs. PSQI).

Response: For test-retest reliability, we compared the first and second measurements of the ASSQ-SDS and ASSQ-Chronotype using ICC analysis. Regarding the convergent validity, we investigated the correlation of the ASSQ-SDS and ASSQ-Chronotype with the PSQI questionnaire. We hope that our response addresses your concern.

Besides these changes, the authors should also provide more information to improve the robustness of the analysis. First, the authors should present the required sample size for the selected reliability coefficients. You can determine the sample size based on one reliability coefficient (e.g., Cronbach’s alpha or ICC). For example, if you use Cronbach’s alpha, you must indicate the minimum acceptable Cronbach’s alpha, expected Cronbach’s alpha, significance level, statistical power, number of items, and expected dropout rate. On the other hand, if you use ICC, you must indicate the minimum acceptable ICC, expected ICC, significance level, statistical power, number of raters or repetitions per subject, and expected dropout rate. Please, follow this article to determine the sample size: https://doi.org/10.21315/eimj2018.10.3.8.

Response: Thank you for asking this question. We followed the published instruction by Boateng et al. (2018) to estimate the sample size for our study. According to their study, the rule of thumb is to have at least 10 participants for each scale item (an ideal ratio is 10:1), with a larger sample size (or respondent:item ratio) is preferable. In our study, the ASSQ-SDS consists of five questions, while the ASSQ-Chronotype consists of four questions. Therefore, we planned to include at least 50 participants, and we successfully recruited 90 athletes by the end of the data collection period.

Second, Bland-Altman plots with 95% limits of agreement (LOA) should be presented to demonstrate the systematic bias/differences between results. You can follow the article “Measuring agreement in method comparison studies” by Bland and Altman (DOI: 10.1177/096228029900800204).

Response: Thank you for providing this insightful feedback. Bland-Altman plots have been added to the revised manuscript.

LL119-121: For SEM, you should first calculate the standard deviation of the differences between scores and then divide by √2 (see this book https://doi.org/10.1017/CBO9780511996214 to understand the calculations and why the authors discourage using the formula SEM = SD x √(1-ICC)). To calculate SDC, multiply the SEM by 1.96 and √2, as you already did. I also suggest presenting SEM and SDC’s relative (%) values.

Response: We appreciate the reviewer’s constructive feedback. The SEM was re-calculated using the suggested formula provided by the reviewer. The new SEM and SDC values for the ASSQ-SDS and ASSQ-Chronotype have been updated in the results section of the revised manuscript. Moreover, the SEM formula has been updated in the method section.

5- Results (Mohammed)

LL141: Please, complement the text with a Table ( characteristics of the participants). More information should be added, including sex, age, body height, body mass, and sport practiced.

   Response: Thank you for your response. We have provided a table of sample characteristics.

After “3.4. Internal consistency”, please report the systematic bias between the first and second administrations for the ASSQ-SDS and ASSQ-chronotypes, and the differences between ASSQ-SDS vs. PSQI and ASSQ-Chronotype vs. PSQI.

Response: The systematic bias between the first and second administrations for the ASSQ-SDS and ASSQ-chronotypes has been addressed by providing Bland-Altman plots. As for the second part of the comment, we are not sure what the reviewer means by “the differences between ASSQ-SDS vs. PSQI and ASSQ-Chronotype vs. PSQI.”

LL160: The authors should also report the inter-reliability analysis (ASSQ-SDS vs. PSQI and ASSQ-Chronotype vs. PSQI).

 Response: Please refer to our response to your comment about the inter-reliability.

6- Discussion (Bader)

  1. “Main findings”: succinctly describe the results;

Response: Thank you for providing this insightful feedback. The discussion has been rewritten to address your valuable comments.

  1. “Internal consistency and systematic bias”: discuss and compare the results (Cronbach’s alpha and Bland-Altman plots) with the available literature;

 Response: Thank you for providing this insightful feedback. The discussion has been rewritten to address your valuable comments.

  • “Test-retest reliability”: discuss and compare the results (intra and inter-reliability) with the available literature;

 Response: Thank you for providing this insightful feedback. The discussion has been rewritten to address your valuable comments.

  1. “Construct validity”: discuss and compare the results (correlations) with the available literature;

Response: Thank you for providing this insightful feedback. The discussion has been rewritten to address your valuable comments.

  1. “Clinical implications”: describe the clinical and sports implications;

 Response: Thank you for providing this insightful feedback. The discussion has been rewritten to address your valuable comments.

  1. “Strengths and limitations of the study”: present the main strengths and limitations of the study with a focus on future research.

 Response: Thank you for providing this insightful feedback. The discussion has been rewritten to address your valuable comments

Reviewer 5 Report

The paper is very interesting and well described. I congratulate the authors for their work.

There is just some minor concerns that I want to address to the authors:

The methods used for the validation seems to be appropriate. The results of the validation are clearly reported. However, in addition to Table 1, a chart could be useful to better understand the result you obtained in your sample (you can use stacked bar charts based on percentage for example). You could report the data in two charts, respectively for test and retest results, for a better interpretation of the significant differences you obtained and to evaluate whether some items are more “modifiable” than others.

Throughout the manuscript you used “ASSQ-AR” and “ASSQ-Ar”. It would be better to always use the same version.

Author Response

Thank you for allowing us to submit a revised draft of our manuscript titled “Translation and Validation of the Arabic Version of the Athlete Sleep Screening Questionnaire.” We appreciate the time and effort that you and the reviewers have dedicated to providing your valuable feedback on our manuscript. We are grateful to the reviewers for their insightful comments on our paper. We have been able to incorporate changes to reflect most of the suggestions provided by the reviewers. We have highlighted the changes within the manuscript.

Here is a point-by-point response to the reviewers’ comments and concerns.

in addition to Table 1, a chart could be useful to better understand the result you obtained in your sample (you can use stacked bar charts based on percentage for example). You could report the data in two charts, respectively for test and retest results, for a better interpretation of the significant differences you obtained and to evaluate whether some items are more “modifiable” than others.

 Response:
Thank you for providing this insightful feedback. Bland-Altman plots have been added to the revised manuscript.

Throughout the manuscript you used “ASSQ-AR” and “ASSQ-Ar”. It would be better to always use the same version.

Response: Thank you for your comment. The term “ ASSQ-Ar” was used throughout the manuscript now.

Round 2

Reviewer 4 Report

Dear authors,

I want to congratulate you for having revised the manuscript as requested. Although you addressed most of my comments, you did not include the sample size estimation in the manuscript. In addition, English grammar and spelling still need to be improved. After these revisions, I believe the manuscript will be ready for publication.

Best regards.

Author Response

Dear Reviewer, 

Thank you for you valuable feedback and your time. Here is a point by point responses to your comments.

  • You did not include the sample size estimation in the manuscript. In addition,
  • Thank you for your comment. we have added a paragraph under the participants subsection explaining the sample size needed.
  • English grammar and spelling still need to be improved. 
  • thank you for your feedback. a full editing and proofreading was preformed to the whole manuscript.